



# Scale separation for gravity wave analysis from 3D temperature observations in the MLT region

Björn Linder[1], Peter Preusse[2], Qiuyu Chen[2], Ole Martin Christensen[1], Lukas Krasauskas[1], Linda Megner[1], Manfred Ern[2], and Jörg Gumbel[1]

[1]Department of Meteorology, Stockholm University, Stockholm, Sweden
[2]Institute of Energy and Climate Research (IEK-7: Stratosphere), Forschungszentrum Jülich GmbH, Jülich, Germany

**Correspondence:** Björn Linder (bjorn.linder@misu.su.se)

**Abstract.** MATS (Mesospheric Airglow/Aerosol, Tomography & Spectroscopy) is a Swedish satellite designed to investigate atmospheric dynamics in the Mesosphere and Lower Thermosphere (MLT). From observing structures in noctilucent clouds over polar regions, and oxygen A-band emissions globally, MATS will supply the research community with properties of the MLT atmospheric wave field. Individual A-band images taken by the MATS main instrument, a 6-channel limb imager, are through tomography and spectroscopy turned into three-dimensional temperature fields in which the wave structures are embedded. To identify wave properties, in particular the gravity wave momentum flux, from the temperature field, smaller-scale perturbations, associated with the targeted waves, must be separated from large-scale background variations by a method of scale separation. This paper investigates the possibilities of employing a simple method based on smoothing polynomials to separate the smaller and larger scales. By using synthetic tomography data based on the HIAMCM (High Altitude Mechanistic Circulation Model) we demonstrate that smoothing polynomials can be applied to MLT temperatures to obtain fields corresponding to a global scale separation at zonal wavenumber 18. The simplicity of the method makes it a promising candidate for studying wave dynamics in the MATS temperature fields.

## 1 Introduction

Gravity waves (GWs) are important features of the dynamics of Earth's atmosphere. Generation mechanisms are numerous, for example, flow over topography, convection and geostrophic adjustment processes (Fritts and Alexander, 2003). GWs may propagate over long distances from their sites of generation, redistributing energy and momentum geographically as well as vertically. As the waves break, these dynamical quantities are transferred to the surrounding medium, affecting its temperature and motion. On a global scale, the redistribution of energy and momentum due to GWs affects circulation patterns in the middle atmosphere, generating the pole-to-pole circulation in the mesosphere (Lindzen, 1981; Holton, 1982). Breaking waves may through their body forces generate secondary waves which may, in turn, generate tertiary waves (Vadas and Fritts, 2002; Vadas and Becker, 2019). As the waves perturb density, pressure and temperature their presence leaves structures in noctilucent clouds (Pautet et al., 2011; Taylor et al., 2011) and atmospheric airglow (Nakamura et al., 1999; Medeiros et al., 2004).





MATS (Mesospheric Airglow/Aerosol, Tomography & Spectroscopy) is a Swedish satellite funded by the Swedish National Space Agency (Gumbel et al., 2020). MATS was launched on November 4, 2022, from Mahia Peninsula, New Zealand, with a scientific objective to identify and quantify atmospheric wave activity in the MLT. To accomplish its mission, MATS carries a 6-channel limb imager with two main IR channels centred on 762 nm and 763 nm, targeting airglow emissions in the $O_2(^1\Sigma_g^+ - ^3\Sigma_g^-)$ Atmospheric band ("A-band"). The channels produce 2D images of the atmospheric limb. Through tomography and spectroscopy, individual images taken along the orbit track are combined to produce three-dimensional temperature fields (Gumbel et al., 2020). From these 3D fields, gravity wave properties will be derived, including for example global maps of wave vectors, wave amplitudes, and the vertical flux of horizontal gravity wave pseudomomentum.

The first step to identify wave properties from temperature measurements is to isolate the wave-induced disturbances $T'$, and the corresponding background $\overline{T}$, from the full temperature $T$. For this, a background removal scheme is applied to the data. Following linear wave theory, the temperature measurements made by MATS can be considered to consist of a superposition of the large-scale structures, the temperature background $\overline{T}$, and the smaller-scale structures due to the GWs, i.e. the temperature residuals $T'$, as

$$T(x,y,z,t) = \overline{T}(x,y,z,t) + T'(x,y,z,t). \tag{1}$$

The coordinates of Equation 1 refer to the geometry of the MATS tomography output, where $x$ refers to the along-track distance, $y$ the across-track distance, $z$ the height and $t$ the time. Different methods for background removal have been discussed by Sakib and Yigit (2022). The separation of scales is typically made based on research interests and/or measurement limitations. Ideally, all of the waves that are small enough to be identified in the MATS data geometry should be included in the residual field, and larger structures should be included in the temperature background.

The distinction between scales are typically made based on the zonal wavenumber (ZWN),

$$k = \frac{2\pi R \cos\varphi}{\lambda}, \tag{2}$$

where $R$ is the radius of the Earth, $\lambda$ is the wavelength in the zonal direction and $\varphi$ is the latitude. One method to perform scale separation is fitting waves in the zonal direction and cutting at a specified zonal wavenumber. This method can be combined with polynomial smoothing in the meridional direction (Ern et al., 2018; Strube et al., 2020). Low wavenumbers are associated with the background, and high wavenumbers with the GW fields. This method has the advantage of a clear cut in the horizontal wave spectrum and no degradation in the vertical wave spectrum of the temperature residuals. It is easily performed on model data on synoptic time scales but becomes very demanding for satellite data in the MLT where tides and other fast planetary wave modes also need to be included in the global wave fits (Ern et al., 2018).

An alternative to the global fitting, in particular for 3D observational data, is the application of local background estimates, which generally are based on smoothing of the observed fields (e.g. AIRS, Hoffmann and Alexander (2010) and GLORIA, Krisch et al. (2017)). These methods have the advantage of relying only on local data, but the cut in the wave spectrum is not sharp and depends on the particular smoothing parameters. These parameters thus have to be selected so the background estimate does not capture a significant part of the targeted wave spectrum. The aim of this study is thus to use synthetic





satellite data, generated from general circulation model output, to test the local scale separation and to use the global scale separation as a reference. For this, the global scale separation is first performed on the global model fields. Based on this, both full temperatures and reference temperature residuals ($T'_G$) are sampled to the MATS observation geometry. A second set of temperature residuals ($T'_L$) is generated via local scale separation. Both $T'_G$ and $T'_L$ are then analyzed with respect to gravity waves and the results are compared. In this way, the local scale separation can be optimized and validated against the reference global scale separation.

The cut in the zonal wave spectrum of the global scale separation should be at a wavenumber that clearly separates out the waves of interest, in our case the smaller scale GWs, from the other type of disturbances that constitute the background. Strube et al. (2020) showed that the ZWN $\geq 18$ residual fields capture the waves carrying the significant GWMF in the upper troposphere and lower stratosphere. Using the same residuals and additional meridional smoothing, Chen et al. (2022) successfully captured the ZWN $\geq 18$ wave dynamics in the MLT using the wave analysis tool S3D (Lehmann et al., 2012). Based on these studies, we investigate the possibility of deriving the MLT wave dynamics from MATS data by resampling residual and background fields decomposed at ZWN 18. The atmospheric data in the study is taken from the atmospheric model HIAMCM (Section 2.2) and the fields are resampled to the expected data geometry of MATS using an orbit simulator (Section 2.3). The full temperature field $T(x, y, z)$ is separated using a local background removal, a Savitzky-Golay (SG) polynomial smoothing filter (Section 2.5). The wave properties are retrieved using the wave analysis tool S3D (Section 2.4) and the local removal is assessed by comparing the obtained wave properties (wavenumbers, momentum flux) with those derived from the reference global scale separation. Finally, a more realistic simulated tomography product is generated by applying averaging kernels (AVKs) and Gaussian noise (Section 2.6) to the full temperature field $T(x, y, z)$, and the SG filter is thus evaluated under more lifelike conditions. The complete analysis chain is described in Section 3. In summary, we ask ourselves

1. Can MLT wave dynamics be derived from the residuals $T'(x, y, z)$ in the MATS data geometry using S3D?

2. Can we isolate the same wave dynamics by applying a local scale separation based on polynomial smoothing to the full temperature field $T(x, y, z)$?

3. Does the above also work for temperature data that have been degraded in the tomographic retrieval process?

## 2 Theory, models and methods

### 2.1 Vertical flux of horizontal gravity wave momentum

Vertical flux of horizontal gravity wave momentum is defined as

$$(F_x, F_y) = \overline{\rho}(\langle u'w' \rangle, \langle v'w' \rangle), \tag{3}$$

where $F_x$ and $F_y$ are the zonal and meridional components and $\overline{\rho}$ is the background density (Fritts and Alexander, 2003). The angle brackets around the wind residuals $u', v', w'$ indicate that they are averages over a spatial region that largely includes the



wavelengths of the waves considered. Ern et al. (2004) showed that Equation 3 can be written in an alternative way,

$$(F_x, F_y) = \frac{1}{2}\overline{\rho}\left(\frac{g}{N}\right)^2 \left(\frac{\hat{T}}{\overline{\overline{T}}}\right)^2 \frac{(k,l)}{m}\left(1 - \frac{f^2}{\omega^2}\right)^{-1},$$
(4)

where g is the gravitational acceleration, $N$ is the buoyancy frequency, $\hat{T}$ and $\overline{T}$ are the temperature amplitude and the background temperature, respectively. $k, l, m$ are the horizontal and vertical wavenumbers, $f$ is the Coriolis parameter and $\omega$ is the intrinsic frequency. Equation 4 is simplified by excluding two correction terms which are described by Ern et al. (2017). Note that Equation 3 calculates the momentum flux directly from the basic wind field while the representation in Equation 4 explicitly depends on wave parameters. Investigations into how well gravity wave parameters were retrieved can thus be done by applying both expressions and comparing the results. The rightmost term is the conversion factor between GW momentum flux and the GW *pseudo*momentum flux (Fritts and Alexander, 2003). The vertical flux of horizontal *pseudo*momentum is an important quantity for propagating atmospheric waves as it is conserved in the absence of dissipation. However, it cannot be expressed without wave properties and is thus not suitable for evaluating the retrieval of waves. Further note that both Equation 3 and Equation 4 contain background properties ($\overline{\rho}$ and $\overline{T}$), highlighting the need to separate the scales correctly.

## 2.2  HIAMCM

The atmospheric data are taken from the HIAMCM (HIgh Altitude Mechanistic Circulation Model). The HIAMCM is a high-resolution, spectral circulation model based on the KMCM (Kühlungsborn Mechanistic Circulation Model) (Becker and Vadas, 2020; Becker et al., 2022). The model was chosen as it explicitly generates atmospheric waves from the surface up to 450 km altitude, with waves represented down to horizontal wavelengths of approximately 200 km, which is smaller than most general circulation models. Because of the turbulence scheme in the model, a sponge layer is not required and the model completely supports the generation of secondary and tertiary waves generated from first principles. The GW length scales that are represented in the model are small enough to be captured by MATS observations, but it is important to note that MATS will also observe waves that are smaller than those that most of today's global models generate. The accurate representation of the larger-scale structures, including the temperature background, makes the HIAMCM data a good candidate for testing background removal. As the properties of the smoothing polynomials are determined based on the large-scale variations of the temperature data, the introduction of additional small-scale structures should not have any substantial effect on the method's ability to capture the dynamics. Scales smaller than those represented in the HIAMCM would mainly affect the residual field. Hence, conclusions drawn concerning the background removal should be valid for the MATS operational analysis independent of the presence or absence of smaller scales. For our analysis, we are using a HIAMCM snapshot from 1st January 2016 at 06:00 UTC without considering any temporal evolution. The global HIAMCM data fields are scale-separated using Fourier decomposition based on zonal wavenumber, with some additional meridional smoothing from rolling third-order polynomials across 5° latitude windows. Motivated by Chen et al. (2022) the scale separation is made at ZWN 18 which results in a dataset that contains large-scale background fields (ZWN < 18) as well as small-scale residual fields (ZWN ≥ 18).



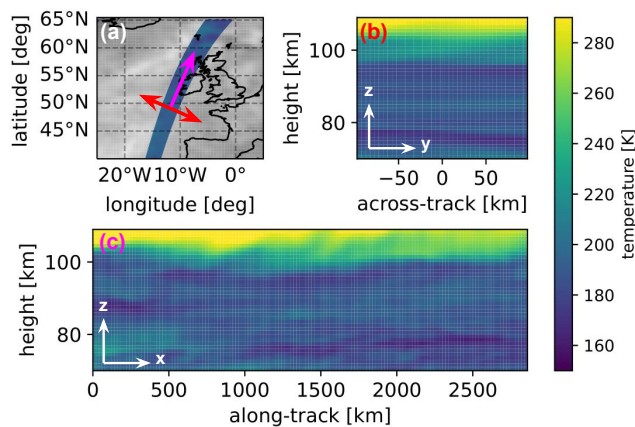

**Figure 1.** Plots showing the HIAMCM output resampled onto the MATS tomography grid. a) The orbit swath with resampled temperature off the coast of the British islands. The arrows illustrate where the slices in the accompanying plots are taken. b) A yz-slice taken in the across-track direction of the orbit swath. c) An xz-slice taken in the along-track direction.

## 2.3 Orbit simulator

A simple orbit simulator for a circular orbit is used to simulate the MATS tomography grid, i.e. a three-dimensional latitude-longitude-height grid, with corresponding, local $(x, y, z)$ kilometre distances. HIAMCM output fields, including temperature, pressure, wind, and density, are interpolated onto this grid. The simulator's calculations are based on a spherical, homogeneous, point-mass Earth without orbit precession. Parameters were determined from preliminary orbit properties before MATS was launched. They vary slightly from the final orbit of MATS, which ended up at 17:30 local solar time (ascending node) and an altitude of 590 km. The output from the MATS tomography may vary in resolution and dimension both geographically and in time, as the settings of the instrument may be controlled based on external factors. However, the resolution and dimension of the temperature grid used in the current analysis are representative of the estimated output from the MATS tomography. The temperature is expected to be obtained between 70 and 110 km, at a vertical resolution of 0.5 km. Across-track, we expect approximately 200 km of data, with a resolution of 5 km. Along-track resolution is expected to be 20 km. As of January 2024, the tomography is still under development and the sampling used in the orbit generator is still a good estimate. In Table 1 the properties used for the orbit simulator are shown. An example of sampled temperature fields can be seen in Fig. 1, where the $(x, y, z)$ directions are illustrated.

## 2.4 S3D

S3D is a wave analysis method based on sinusoidal fits developed by the Stratospheric Group, IEK-7, at the Research Centre Jülich in Germany (Lehmann et al., 2012). Residual temperature fields containing waves are divided into overlapping analysis volumes, and in each volume, waves are identified using an iterative 3D sinusoidal fit technique based on least-square criteria.



**Table 1.** Properties used in the orbit simulator for generating the MATS tomography grid.

| | |
|---|---|
| Orbit inclination | 97.6° |
| Orbit altitude | 560 km |
| Local time (ascending node) | 21:00 |
| Across-track range | 195 km |
| Vertical range | 39 km |
| Size of across-track pixels | 5 km |
| Size of vertical pixels | 0.5 km |
| Sampling distance along track | 20 km |

These analysis volumes are also referred to as cuboids. Once a wave is identified in a cuboid, the wave is subtracted from the data and the sinusoidal fit is repeated, typically three times, so that for each cuboid three dominant waves and their properties are identified. Naturally, S3D cannot identify waves with wavelengths much longer than the cuboid size and the selected cuboid size thus limits the waves that can be detected. To detect a wave, S3D typically requires the cuboid to contain at least one-third of a wavelength in each dimension. It can thus identify wavelengths up to three times the cuboid size (Preusse et al., 2012).

Combining the wave parameters with background parameters the total GWMF can be calculated (Equation 4). GWMF in the MLT can be captured well by applying S3D to temperature residuals obtained from global zonal wavenumber decomposition (Chen et al., 2022). In the same study, analysis volumes of 600 km x 600 km x 20 km and 300 km x 300 km x 15 km were used to retrieve GWMF at altitudes of 75 km and 130 km, respectively. Ideally, cuboids should be large enough to capture all significant wave momentum, but not too large to lose spatial information by mixing wave parameters from a large altitude range.

The dimensions of the MATS tomography output limit the sizes of the S3D analysis volumes that can be used in the operational wave analysis. Specifically, the across-track dimension is the bottleneck, as its range only covers $\sim 200$ km. Because of the large horizontal wavelengths of the waves represented in HIAMCM, cuboid sizes should cover the entire across-track range while keeping a large along-track range. The cuboid sizes are therefore chosen to be 600 km x 190 km x 10 km. Each cuboid will be aligned with the centre of the across-track range of the data, with overlapping cuboids positioned every 100 km in the

along-track direction. Vertically, analysis cuboids are positioned every 5 km between the altitudes of 80 and 100 km. Using these analysis cuboids the wave vectors are identified. These vectors are then used in a so-called refit, where the vertical extent of the cuboids is reduced to 3 km, to accurately determine the wave amplitudes at the centre of each cuboid.



## 2.5 Savitzky-Golay filter

The S3D wave analysis described in the previous section needs the residual temperature field $T'$ as input. It is therefore essen-
tial to carry out the temperature field decomposition described by Equation 1. In our approach, the background temperature
$\overline{T}(x, y, z)$ is estimated by applying a Savitzky-Golay (SG) smoothing filter (Savitzky and Golay, 1964) to the full temperature
field $T(x, y, z)$. The smoothing is applied to each dimension independently; first to the vertical direction, secondly to the along-
track direction and finally to the across-track direction. The temperature residuals are acquired by subtracting the temperature
background from the full temperature.

A Savitzky-Golay filter is similar to a rolling mean filter, but instead of calculating the mean of the data points within the
window, a polynomial of a certain order is fitted. If a zero-degree polynomial is selected the algorithm becomes identical to a
rolling mean filter. For smoothing along the $z$-direction, a polynomial

$$\widetilde{T}(z) = \sum_{i=0}^{p_z} A_i z^i \tag{5}$$

of order $p_z$ is fitted using a least squares criterion. This is done over a smoothing window containing $n_z$ temperature points,
centered on $T(z = z_j)$. The value of the polynomial at the centre point, $\widetilde{T}(z_j)$, is extracted as the background value $\overline{T}(z_j)$.
The whole $\overline{T}(z)$ is retrieved by shifting the smoothing window, i.e. repeating the process above for every $j$. The method can be
tuned by controlling the orders $p_x$, $p_y$ and $p_z$, as well as the smoothing window sizes $n_x$, $n_y$ and $n_z$. Figure 2 shows how the
number of points in the along-track smoothing window controls the shape of the acquired temperature background, for $p_x = 3$.
The temperature data is arbitrarily selected along the satellite track, for illustrative purposes. A shorter window (smaller $n_x$)
results in an estimated background that resembles the original $T$-curve, while a larger window smooths the curve more. In
the latter case, the resulting temperature curve resembles the ZWN < 18 background temperature. Near the data boundaries,
symmetric intervals are not possible. We then use asymmetric intervals, with fewer points between the central point and the
data boundary.

The polynomial orders used in this study are $p_x = 3$, $p_y = 0$ and $p_z = 2$. Means ($p_y = 0$) are taken over all of the across-
track points since the horizontal scales of the background temperature are assumed to be much larger than the across-track
range of the data. ZWN 18 waves correspond to waves with a zonal wavelength of 2200 km at the equator. As the orders
of the polynomials are held constant, an SG filter will be defined by the number of points in the smoothing windows. More
specifically, it will be defined by the number of points in the along-track direction, $n_x$, and by the number of points in the
vertical direction, $n_z$. In the across-track direction, means are taken across all of the points and hence $n_y = 41$ for all cases.

## 2.6 Averaging kernels and noise

Gaussian white noise and averaging kernels (AVKs) are applied to the synthetic satellite data for a more realistic tomogra-
phy output, effectively blurring temperatures horizontally and vertically (Rodgers, 2000). The AVKs are applied as Gaussian



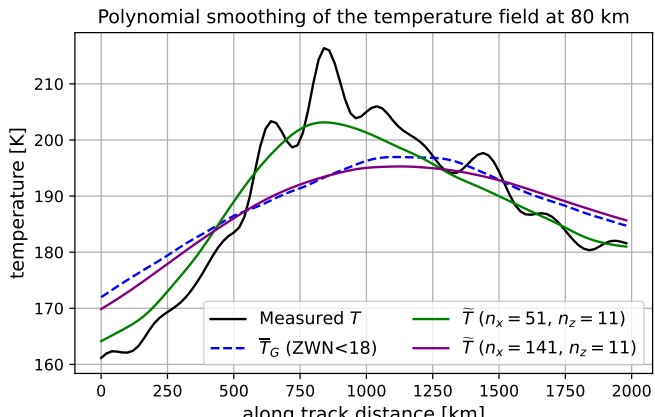

**Figure 2.** Illustration of two different smoothing lengths in the SG filter. The filter with $n_x = 51$ leads to a temperature background resembling the full temperature. When $n_x = 141$, the obtained curve follows the background obtained from zonal wavenumber decomposition. In both of the filters the vertical smoothing is $n_z = 11$.

**Table 2.** The width of averaging kernels used to generate synthetic MATS tomography output

| Name | $\mathrm{FWHM}_x$ | $\mathrm{FWHM}_y$ | $\mathrm{FWHM}_z$ |
|---|---|---|---|
| Averaging kernel 1 (AVK 1) | 60 km | 20 km | 1 km |
| Averaging kernel 2 (AVK 2) | 60 km | 20 km | 2 km |
| Averaging kernel 3 (AVK 3) | 100 km | 20 km | 1 km |

smoothing curves,

$$K(\boldsymbol{x}, \boldsymbol{x}_0) = \exp\left( \sum_{i=1}^{3} -\frac{(x_i - x_{0i})^2}{2\sigma_i^2} \right) \tag{6}$$

where $\boldsymbol{x}$ corresponds to the along-track, across-track and vertical dimensions, $(x_1 = x, x_2 = y, x_3 = z)$, and $\mathbf{x}_0$ are the corresponding central points to which the curve is applied. $\sigma_i$ is the standard deviation along the $i$-th dimension, $(\sigma_1 = \sigma_x, \sigma_2 = \sigma_y, \sigma_3 = \sigma_z)$. We define each kernel based on its full width at half maximum (FWHM) in each dimension, where $\mathrm{FWHM}_i = 2\sqrt{2\ln 2}\sigma_i$. As the final resolution of the MATS tomography output is uncertain, three different AVKs - together covering the expected ranges of measurement resolution, are applied to the idealised tomography output $T(x, y, z)$. This results in three 190 different sets of temperature data. The applied AVKs are described in Table 2.

The Gaussian white noise is added with a standard deviation of 5 K based on uncertainty estimates (Gumbel et al., 2020). Structured noise, such as stray light, that could appear in the raw images taken by MATS, and then propagate into the tomography, is neglected. Such features are assumed to have been removed in the earlier stages of the data processing.





## 3   Analysis chain

The analysis is presented step-by-step in Fig. 3. As introduced in Section 2.1, to evaluate S3D as the operational tool for deriving MLT wave dynamics from MATS tomography output, two datasets of GWMF are produced - one derived from wind residuals and one derived from temperature residuals. For both of these sets, GW-induced perturbations are isolated from the global HIAMCM data using zonal wavenumber decomposition at ZWN 18. These separated fields include wind residuals $u'_G$, $v'_G$, $w'_G$, the background density $\overline{\rho}_G$, the background temperature $\overline{T}_G$, and the residual temperature $T'_G$, where the subscript

$G$ indicates that the field is derived from the global scale separation. As both wind and temperature residuals are present, the GWMF representations described in Section 2.1 can be used equally, i.e. deriving GWMF straight from $u'_G, v'_G, w'_G$ (Equation 3) and deriving GWMF from wave parameters (Equation 4). The GWMF from wind residuals is referred to as Reference (wind residuals) and its derivation is illustrated by the black arrows in Fig. 3. The zonal mean GWMF derived from global wind residuals is considered a "true" reference in that it accurately captures how much momentum is present in the model data, as

discussed by Chen et al. (2022). The GWMF derived from the $\overline{T}_G$ temperature residuals, referred to as Reference (ideal), is obtained by first resampling the separated temperature fields to the geometry of the MATS tomography grid using the orbit generator and then applying S3D. Its derivation is illustrated by the blue arrows in Fig. 3. The comparison between these two sets is presented in Section 4.1.

To investigate the SG filter as a tool for scale separation, the full temperature field $T$ is resampled using the orbit generator.

The filtering is then studied in two alternative approaches. In the first approach, the filter is applied directly to $T(x, y, z)$. This is illustrated by the red arrows in Fig. 3. This dataset will be referred to as the SG filter (ideal) set, as the temperature field is perfectly captured, without any simulated degradation from MATS retrieval. For the scale separation to be successful, we expect the wave spectra in the residual field to agree with the wave spectra of $T'_G(x, y, z)$. The SG filtering is thus evaluated by comparing the SG filter (ideal) set with the GWMF and wavenumber spectra from the Reference (ideal) set. The investigations

into idealised SG filtering are presented in Section 4.2.

In the second approach, more realistic simulated MATS tomography outputs are generated. These are referred to as the SG filter (AVK) sets and their derivation is illustrated by the green arrows in Fig. 3. The realistic sets are put through an extra couple of steps, applying Gaussian noise and AVKs to $T(x, y, z)$ before applying the SG filter. It is important to note that when AVKs have been applied to $T(x, y, z)$, we no longer expect the GW spectra of the temperature field to be the same, as

averaging kernels efficiently smoothen the data set so that small scales disappear. New references are thus required, and these are generated by applying the same AVKs to $T'_G(x, y, z)$. The generation of these new references is illustrated by the orange arrows in Fig. 3 and they are referred to as Reference (AVK). The results are presented in Section 4.4.

As MATS will not be able to measure background density, operationally this will need to be supplied from elsewhere (atmospheric model, complementing measurements). For the current investigations of scale separation, background density

is estimated by resampling the HIAMCM density onto the orbit grid and putting it through the same SG filters as the full temperature. For all of the reference datasets, the background density is retrieved from the global scale separation.

To summarize, the different products described above are compared as follows:





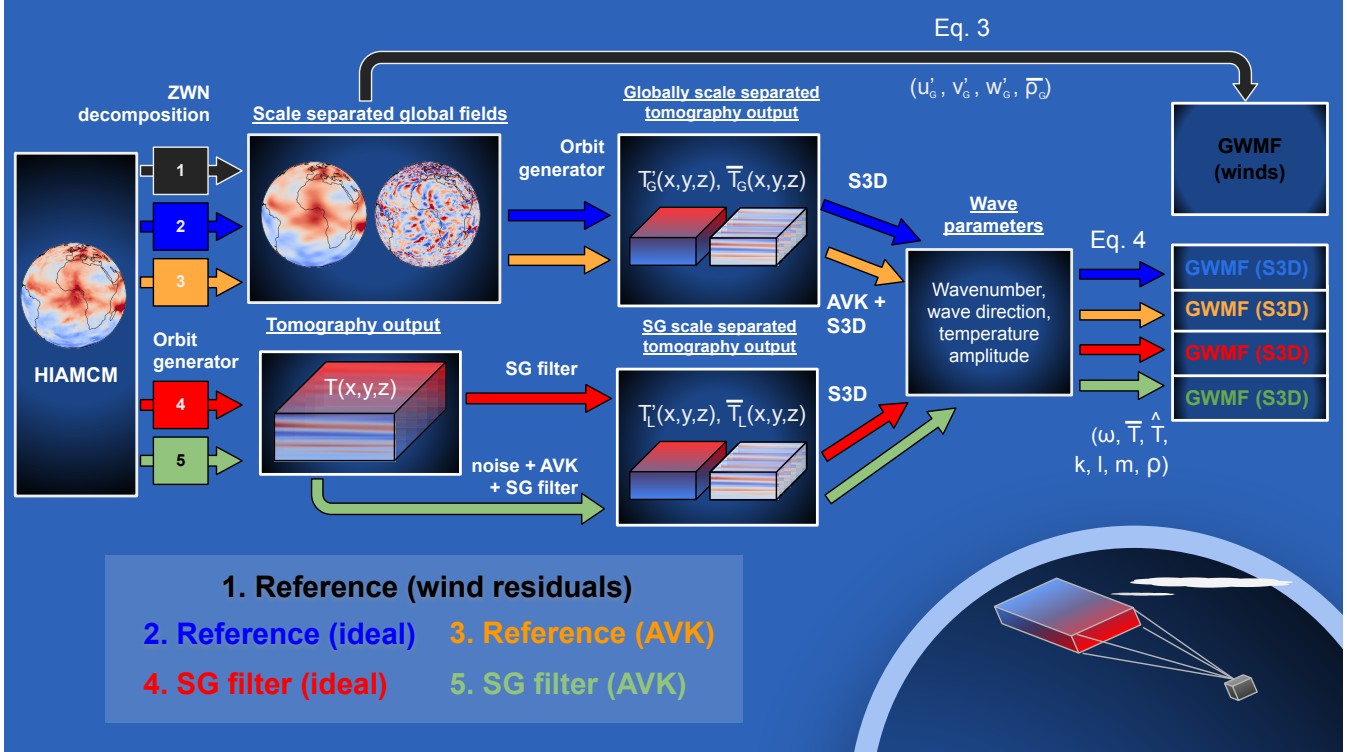

**Figure 3.** A schematic showing the analysis procedure. Five analysis chains are presented. The steps taken to investigate S3D as an operational wave analysis tool are illustrated by the black and blue arrows. The orange, red and green arrows show the steps taken to explore SG filtering as a method for local scale separation in the MATS temperature data.

1. The performance of the S3D wave retrieval is evaluated by comparing sets derived from global scale separation – Reference (ideal) and Reference (wind residuals).

2. SG-filtering of the idealised tomography output is evaluated by comparing SG filter (ideal) to Reference (ideal).

3. SG-filtering of more realistic MATS tomography output is evaluated by comparing SG filter (AVK) to Reference (AVK).

When we compare datasets by studying the zonal mean of the GWMF we apply $5°$ rolling averages over latitude to make data interpretation easier. We also apply 3 km rolling averages in the vertical direction of the GWMF derived from wind residuals. This is done to match the vertical extent of the refitted S3D analysis cuboids.





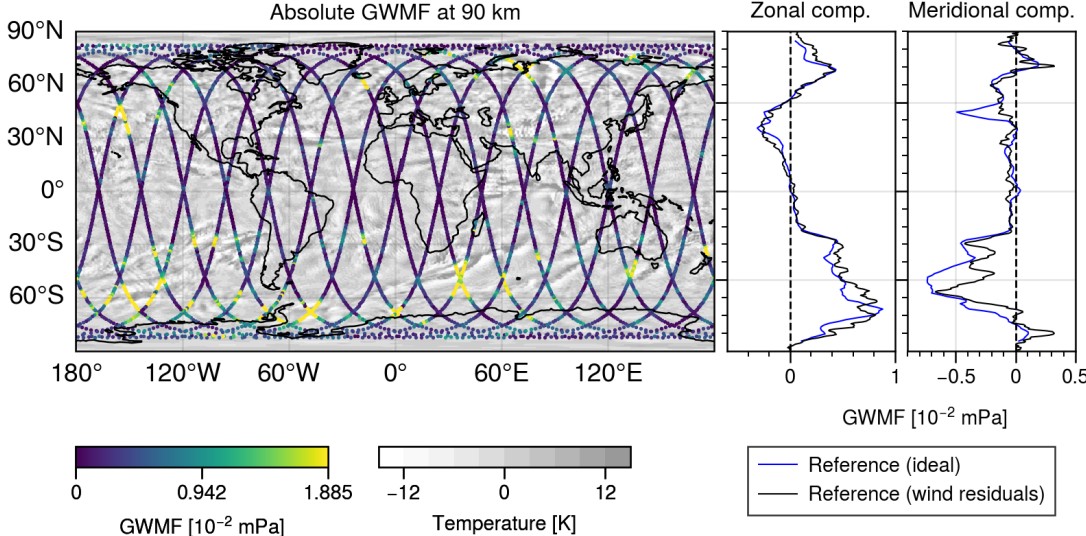

**Figure 4.** GWMF calculated from ZWN $\geq 18$ residuals. In the left plot, the MATS orbit swath shows the absolute GWMF at 90 km derived using S3D. The grey-white colour underneath indicates structures present in the HIAMCM residual temperature field. In the right plots, the zonal mean GWMF from S3D is displayed with GWMF derived from global wind residuals.

## 4 Results and discussion

### 4.1 S3D performance on the MATS tomography grid

To investigate the performance of S3D on MATS data geometry we compare GWMF obtained from the wind residuals $u'_G$, $v'_G$ and $w'_G$ with GWMF obtained using derived wave parameters from $T'_G(x, y, z)$. In Fig. 3 these are referred to as Reference (wind residuals) and Reference (ideal). The left plot of Fig. 4 shows the absolute GWMF at 90 km, derived using S3D on $T'_G(x, y, z)$ along the simulated MATS orbit. Distinctive regions of high GW activity can be identified along the orbit, particularly in the southern hemisphere (SH). The right-hand plots of Fig. 4 display the zonal means of the directional GWMF from wind residuals (black) and S3D (blue). In comparing these two representations, for both components of the GWMF, we find that the S3D analysis cuboids capture momentum with high accuracy at most latitudes, with a few exceptions. In the SH at around 50°S, a region of strong GW activity over the Southern Ocean is observed. In this region, the S3D meridional component is partly overestimated compared to the wind-derived counterpart, attributing the region a stronger flux than what is there. On the other hand, the zonal component is underestimated in a region surrounding 60°S, indicating that the S3D can both over- and underestimate when compared to the wind residuals. The vertical variations in how well S3D captures GWMF are presented along with the results in Section 4.2.

As both the wind residuals and the temperature residuals are derived from the same global scale separation, the difference in GWMF arises mainly from the use of S3D on the temperature residuals (some of the difference can be attributed to the





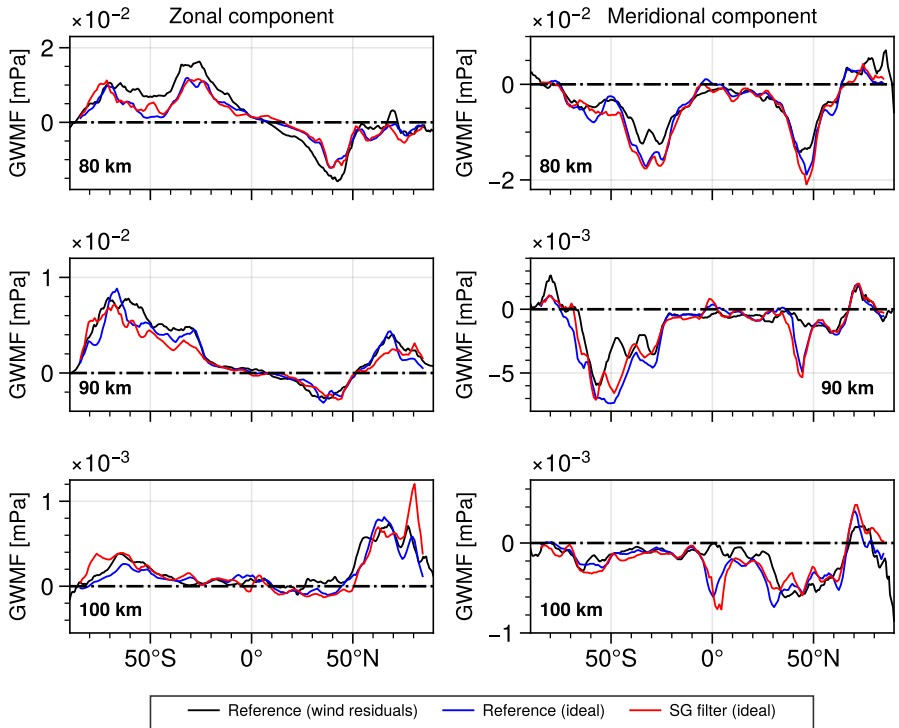

**Figure 5.** The zonal mean of directional GWMF for different altitudes. Every plot shows the GWMF obtained by applying an SG filter to the idealised tomography output, along with the GWMF references derived from ZWN $\geq$ 18 residuals of wind and temperature.

sampling difference between the orbit grid and the global model grid but investigations show that this effect is quite small). So while GWMF derived from the wind residuals (Reference, wind residuals) accurately captures how much momentum is present in the data, when evaluating SG filtering as a method for scale separation, we cannot expect to get any closer to this 'truth' than what the S3D calculations on the $T'_G(x,y,z)$ achieve (Reference, ideal). The latter is therefore used as a reference

when evaluating how well the local SG filtering method separates the scales and not the wind-derived alternative.

## 4.2 Scale separation performance for an idealised product

The GWMF obtained from applying an SG filter to the idealised MATS tomography product (SG filter, ideal) is shown for several altitudes in Fig. 5 (red lines). Together with these results are the GWMF derived from the global scale separation, both the wind residuals (Reference, wind residuals) in black lines and the temperature residuals $T'_G(x,y,z)$ (Reference, ideal)

in blue lines. As mentioned in the previous section, the latter two are in good agreement, in particular in the middle of the altitude range, which was illustrated in Fig. 4. The momentum obtained by applying an SG filter to the temperature field agrees well with the GWMF from the $T'_G(x,y,z)$ reference, for both components and all considered altitudes. This agreement is particularly strong at 80 km, where the curves follow each other closely, throughout the latitudes.



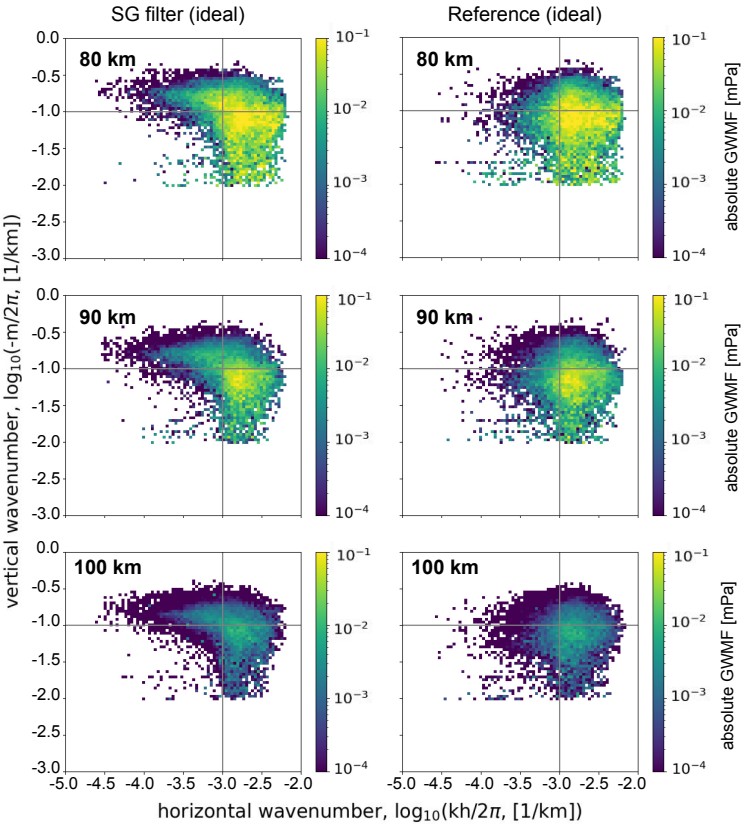

**Figure 6.** Spectra showing absolute GWMF as a function of horizontal and vertical wavenumbers. In the left column, data is derived by applying the SG filter to the MATS tomography output (SG filter, ideal). Data in the right column is derived from the global, ZWN decomposition of the temperature field sampled onto the MATS tomography grid (Reference, ideal).

In Fig. 5, SG filtering has been performed using a smoothing window size of $n_x = 81$ points in the along-track direction and $n_z = 11$ points in the vertical direction. This corresponds to a smoothing window size of 1600 km along-track and 10 km vertically. The successful reproduction of the GWMF in most latitudes confirms a successful separation between temperature residuals and background at ZWN 18. The momentum is overestimated in the zonal components towards the poles, which is especially evident at the 100 km level. In these polar sections of the orbit, the along-track direction of $T(x, y, z)$ is mainly zonal (see Fig. 4). This can interfere with the polynomial smoothing in the filter, as the along-track smoothing has mainly been optimized on the meridional structures. The effect is not as strong at 80 km altitude. The problem of finding a general SG filter that captures horizontal scales equally well across the studied altitude range of 40 km appears difficult.

To retrieve the GWMF correctly we need to ensure that the waves carrying a substantial part of this momentum are retrieved well. In Fig. 6 the wavenumber spectra from the S3D reference (Reference, ideal) and the idealised SG filtering (SG filter, ideal) are shown. The momentum peaks and the surrounding, significant momentum are captured well by S3D from both





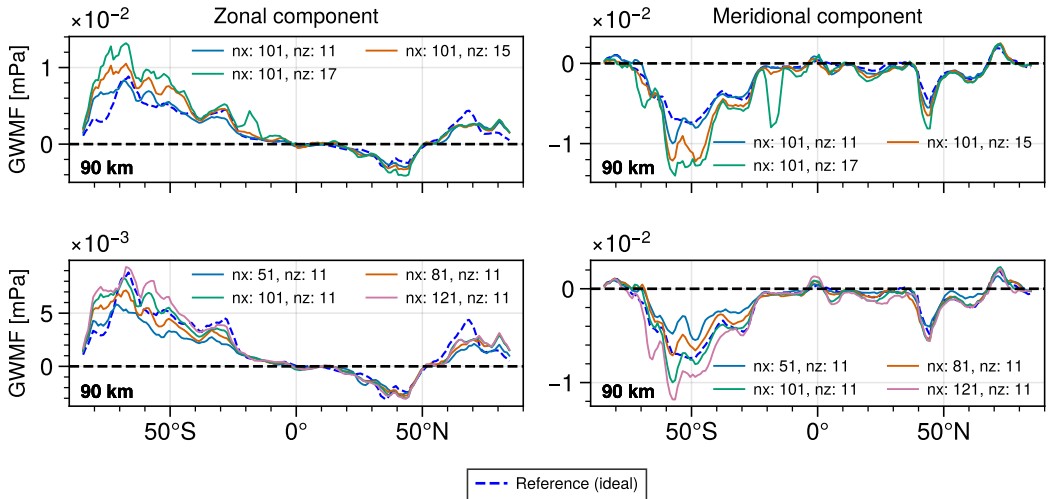

**Figure 7.** Response of the zonal mean GWMF to different smoothing window sizes in the SG filter. The references display GWMF calculated from ZWN $\geq 18$ temperature residuals (Reference, ideal). Each filter is defined by its number of along-track points $n_x$ and vertical points $n_z$.

the $T'_G(x, y, z)$ and the SG-filtered temperature fields. The spectra obtained through the SG filter contain additional wavelike structures that are not in the wave spectra of the reference, indicated by the elongated 'tails' in the left-hand plots. These waves have small horizontal wavenumbers and large vertical wavenumbers, indicating they are large-scale structures that get interpreted as waves in the GW spectra. To capture the dynamics, it is important that the wave spectrum, especially where the momentum is found, is retrieved without losses. Noting the logarithmic colour scale it is clear that the missing, as well as the

introduced, waves only carry a small amount of the momentum, which is promising for the separation method.

## 4.3   Sensitivity and response

The vertical and horizontal smoothing of the SG filter affects the wave spectra and GWMF differently, with a larger sensitivity to changes in the vertical smoothing. In Fig. 7 the zonal mean GWMF response to smoothing window changes is shown from 90 km altitude. The results can be applied to all altitudes studied. Figure 7 illustrates that fewer points in the smoothing

window typically lead to less GWMF. For a shorter interval, the estimated temperature background will include more of the GW-induced perturbations, thus leaving less variation for the wave analysis. Vertical smoothing has a substantial effect on the data, a small reduction in the number of points in the vertical smoothing interval significantly affects the momentum flux. This is not surprising as the vertical resolution is close to the length scales of the vertical wavelengths in the data. Consequently, small changes can cut off significant amounts of momentum.

In general, it should be noted that sensitivity depends on latitude. In the northern hemisphere, the response to changes in the smoothing window size is much smaller. As discussed in Chen et al. (2022), in the southern hemisphere MLT of the considered





HIAMCM snapshot, the vertical gradient of GWMF is strong around 90 km, indicative of strong vertical wind gradients and dissipating gravity waves. Nonlinear wave dynamics and short vertical wavelengths make sinusoidal fitting more difficult and thus more responsive to changes in the residual temperature field. As noted in connection with Fig. 6, momentum may get
artificially enhanced in regions due to smoothing - where large-scale structures get interpreted as GWs.

## 4.4 Scale separation performance for realistic products

We now apply the SG filter from Section 4.2 to the more realistic tomography products - temperature fields subjected to Gaussian noise and AVKs. In Fig. 8 the GWMF derived from the filtering of these fields are displayed (SG filter, AVK) as solid lines, along with the associated references (Reference, AVK) as dashed lines. The type of AVK applied is indicated by
300 the colour of the line. The orange lines show the GWMF from Section 4.2, derived from the idealised tomography product (SG filter, ideal) and its reference (Reference, ideal). Figures 8.b and 8.d display the ratios between the GWMF references and the GWMF obtained through SG filtering. Occasionally the ratios diverge, not because the two zonal means differ much in absolute terms, but because the momentum flux derived from applying the SG filter approaches zero. These segments are highlighted in red and should be ignored.

As explained in Section 3, the reference zonal means are computed by applying the same AVKs to the residual temperatures $T'_G(x, y, z)$ from global scale separation. By considering the references of Fig. 8, it is clear that the AVKs remove waves in the GW spectra and consequently momentum in the zonal mean. For example, in the zonal component (Fig. 8.a the GWMF drops near 70°S, compared to Reference (ideal), for all kernels applied. The largest reduction is seen when the kernel with the largest vertical averaging is applied, namely AVK 2 with $FWHM_z = 2$ km. This illustrates that a loss in vertical resolution of the
measurements has a large effect on the observable waves. Similar conclusions can be drawn from the meridional component of the momentum flux, by considering the reference zonal means in Fig. 8.c.

Optimized for the idealised tomography product, the SG filter is the most effective when no AVKs are applied. In the southern hemisphere, the ratios displayed in Fig. 8.b and Fig. 8.d show that without data degradation, the reference is within a factor 1.5 of the momentum obtained through SG filtering (orange lines). In the northern hemisphere, the results are slightly
worse and the zonal component from SG filtering is partially underestimated, down to half of the reference. Very little GWMF is identified near the equator and toward the poles, as mentioned in Section 4.2, the momentum is artificially enhanced.

The performance of the SG filter drops when it is applied to temperature fields subjected to AVKs. This is seen from the ratios, which are further from unity than the counterparts derived from idealised tomography output. In general, the GWMF derived from the filters stays within a factor 2 of their corresponding references. In the zonal component of the GWMF, the
320 scale separation seems equally successful for all AVKs, with some latitudinal variation. In the meridional component, the scale separation of the tomography product where AVK 2 was applied was the least successful. This kernel corresponds to the largest loss of vertical resolution. The explanation as to why the performance of the SG filter is affected by the kernels is that the AVKs generate a field that is generally smoother. The smoothing windows defined in the filter now capture a background that incorrectly includes some of the larger-scale waves. This is particularly important in the vertical, where the FWHM of
325 the kernels is closer to the length scales of the structures. From Fig. 8 it is clear that the difference in along-track averaging

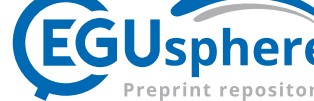

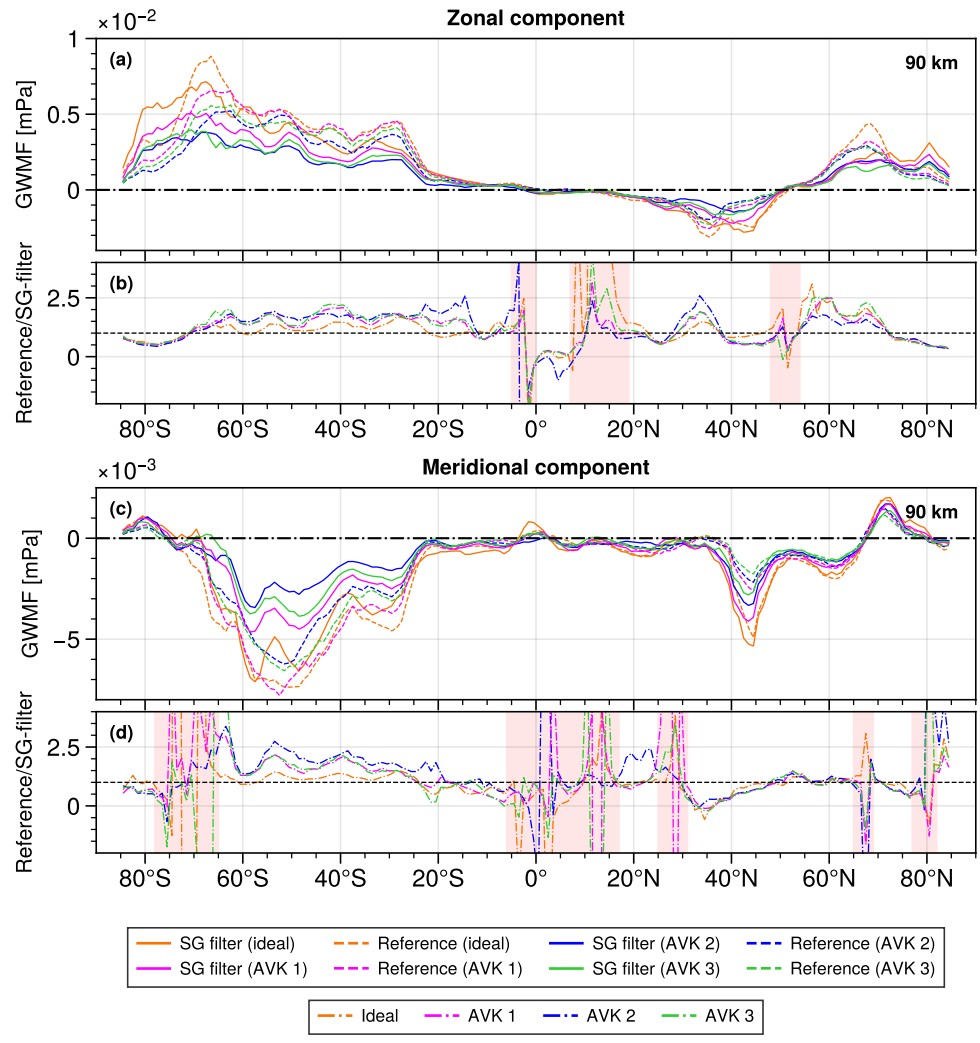

**Figure 8.** GWMF at 90 km derived from temperature fields subjected to noise and AVKs, along with corresponding references. The ratios between the two are also illustrated. For segments where the GWMF derived using the SG filter approaches zero, the ratio is highlighted in red. **AVK 1**: $\mathrm{FWHM}_x = 60$ km, $\mathrm{FWHM}_y = 20$ km and $\mathrm{FWHM}_z = 1$ km, **AVK 2**: $\mathrm{FWHM}_x = 60$ km, $\mathrm{FWHM}_y = 20$ km and $\mathrm{FWHM}_z = 2$ km, **AVK 3**: $\mathrm{FWHM}_x = 100$ km, $\mathrm{FWHM}_y = 20$ km and $\mathrm{FWHM}_z = 1$ km.

between AVK 1, which has $\mathrm{FWHM}_x = 60$ km and AVK 3, with $\mathrm{FWHM}_x = 100$ km, has little effect on the scale separation. Compensation for the smoother temperature field might be possible by revisiting the smoothing window sizes in the filter. This should be done once the experimental AVKs of the MATS temperature retrieval have been determined. The applied noise may explain some of the lost GWMF but typically S3D performs well on noisy data. The results shown in Fig. 8 are taken from





90 km, the middle of the vertical extent of MATS three-dimensional data, but the same conclusions can be drawn from the analysis at 80 km and 100 km.

## 5 Conclusions

In this study, we have shown that the SG filter, a local scale separation based on smoothing polynomials, combined with wave retrieval using S3D, is a suitable operational method for doing wave analysis on MATS temperature measurements. S3D

can characterise, and correctly recreate the vertical flux of horizontal gravity wave momentum in both meridional and zonal directions, from GWs with wavelengths down to $\sim 200$ km, the lower horizontal wavelength limit of the HIAMCM. The SG filter, in all of its simplicity, is a good first candidate for the operational background removal method for MATS, in that it manages to separate the temperature residuals from the temperature background on scales equivalent to ZWN 18 structures. The application of AVKs to the measurements, simulating variations in instrument resolution, reduces the performance of

the scale separation, but the combination of SG filtering and S3D wave analysis should allow the GWMF to be retrieved within a factor of $\sim 2$. Evaluating the methods this way, by comparing the output to a reference, allows us to find well-suited smoothing parameters, a result that only can be made in an ideal model world with full fields and wind data available. Specifically, smoothing window sizes of $n_x = 81$ points in the along-track direction and $n_z = 11$ points in the vertical direction (corresponding to 1600 km along-track and 10 km in the vertical) capture the dynamics well.

The results are a conservative estimate in the sense that in the real MATS measurements, we expect a large fraction of the observed waves to be of smaller horizontal wavelength than what can be observed in the HIAMCM. This will lead to a clearer distinction between waves that carry the majority of the GWMF and the larger scales of the background structures, affecting the scale separation positively. At the same time, the temporal sampling will be different, only collecting data at 17:30 and 05:30 (at the equator) due to the sun-synchronous orbit of MATS. This might affect the structure of the large-scale background

caught in the measurements. As the AVK is a common diagnostic output from retrieval procedures (e.g. Ungermann et al. (2010)) further studies into both observational filter and S3D settings will continue throughout the tomography development, as more information will become available about the large-scale temperatures found in the MATS data.

As MATS observes the three-dimensional temperature fields of the MLT, a remaining matter is the acquisition of additional background fields. Background fields of interest for a deeper wave analysis include atmospheric density, required for the

GWMF calculations made throughout this study, and background winds, required to perform ray tracing. These additional background fields are not measured by MATS but will need to be incorporated through collaboration with other projects.

*Code availability.* The software code is available on request.



*Author contributions.* Björn Linder: Formal analysis, Writing, Visualisation, Peter Preusse: Conceptualization, Methodology, Software, Supervision, Qiuyu Chen: Software, Investigation, Ole Martin Christensen: Resources, Software, Lukas Krasauskas: Conceptualization, Linda Megner: Supervision, Writing, Jörg Gumbel: Supervision, Manfred Ern: Conceptualization.

*Competing interests.* Jörg Gumbel is a member of the editorial board of Atmospheric Measurement Techniques.

*Acknowledgements.* B. Linder, O. M. Christensen, L. Krasauskas, L. Megner and J. Gumbel received funding from the Swedish National Space Agency. We would like to acknowledge the work of Donal Murtagh, Jacek Stegman, Jonas Hedin, Nickolay Ivchenko and Joachim Dillner who are involved in the MATS satellite mission but not directly involved in this study. We thank Erich Becker who supplied the HIAMCM data.



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
