# Peer review of "Scale separation for gravity wave analysis from 3D temperature observations in the MLT region"

_EGUsphere, 2024_

## Author Comment (AC1)

**Author response to Referee #1**

We would like to thank referee #1 for the valuable comments. We have included the comments one by one, in bold text, along with our answers. Lines in the answers refer to the original manuscript. The blue colour indicates text added in the revised manuscript.

- **The analysis focused on cuts on zonal wavenumber, Savitzky-Golay filtering and S3D transforms. Other methods that are also applied to observational data in order to analyze gravity waves like Butterworth filters, wavelet transforms or modal decompositions were not mentioned.**

    It is a good suggestion to mention some examples of other methods. Referee #2 suggested mentioning S-transform. In the revised manuscript we introduce Butterworth and Savitzky-Golay filtering as two examples of scale separation, as well as wavelet transform (in the form of the S-transform) and S3D as two tools for wave analysis on 3D temperature observations. As far as we understand modal decomposition requires "full" model data and would not be applicable (or difficult) for satellite observations. In the updated manuscript we mention the two scale-separation methods and wave analysis tools in the introduction:

    L57: Based on this, both full temperatures and reference temperature residuals (T'$_G$) are sampled to the MATS observation geometry. A second set of temperature residuals (T'$_L$) is generated via local scale separation. The local scale separation in this study is made using a Savitzky-Golay filter (Savitzky and Golay, 1964). An alternative method could be to apply Butterworth filters (Butterworth,1930). Based on a comparison between these separation methods with 3D temperature data we expect the results to be similar (Krisch et al., 2020). Both T'$_G$ and T'$_L$ are then analyzed with respect to gravity waves and the results are compared. In this way, the local scale separation can be optimized and validated against the reference global scale separation. The residuals are analysed using a wave analysis tool. For 3D temperature observations, one alternative is to use the continuous wavelet transform in the form of the S-transform (Wright et al., 2017; Hindley et al., 2019). In this study, we use S3D (Lehmann et al., 2012), a computationally cheap method based on sinusoidal fits.

- **It may be interesting, if feasible, to discuss in more detail one specific wave event when the performance was especially bad, in order to get a feeling on what type of waves would not be well captured. For example the differences in the meridional component at 50 deg N are large, maybe due to a wave with short vertical wavelength?**

    We agree that some further analysis of why local disagreement between S3D and the reference occurs is beneficial for the article. In the revised manuscript we add a discussion to Sect. 4.1 (S3D performance on the MATS tomography grid) regarding the disagreement at 50 deg N:

    L249: As both the wind residuals and the temperature residuals are derived from the same global scale separation, the difference in GWMF arises mainly from the use of S3D on the temperature residuals (some of the difference can be attributed to the sampling difference between the orbit grid and the global model grid but investigations show that this effect is quite small). At 50N the meridional component of the zonal mean GWMF is overestimated by S3D. The specific region from which the discrepancy arises is the area of high GWMF in

the northern Pacific, just south of Alaska, as seen in the map of Fig. 4. Due to the orbit sampling, this area has a large contribution to the zonal mean and if excluded the discrepancy disappears. To understand why the disagreement occurs we visually compared the residual field with the identified waves (not shown). The region in question is characterised by waves with large vertical wavelengths, short meridional wavelengths and large amplitudes, all contributing to large GWMF. The large vertical wavelengths appear to be overestimated by S3D, which in turn leads to an overestimation of the momentum flux. Naturally, it is easier to characterise waves that are smaller than the cuboid and it is not surprising that an area with these large vertical waves is challenging.

- **Is the used HIAMCM snapshot considered representative? In a sense that different snapshots are comparable in covered gravity wave events and variability, such that it does not seem necessary to evaluate different snapshots?**

At the time of the study, we were considering performing tests on a larger set of HIAMCM data but this was abandoned as we could not obtain this dataset. As we only have one snapshot, it is indeed hard to know exactly how representative it is. However, as references and the parameters we derive using local scale separation are from the same snapshot, we are not critically dependent on this. We have added a short discussion regarding this in the conclusions of the revised manuscript:

L345: This study was performed on a HIAMCM snapshot from 1st January 2016. For investigating if the method works, a global snapshot from an arbitrary day should be reasonable as it should include regions where analysis might be harder (for example in the vicinity of wave-breaking) and easier (well-defined wave structures). Nevertheless, for future studies, a larger dataset would be preferable.

- **307 missing closing bracket after (Fig.8a**

This has been corrected in the updated manuscript.

**References**

Butterworth, S.: On the Theory of Filter Amplifiers, Wireless Engineer, 7, 1930.

Hindley, N. P., Wright, C. J., Smith, N. D., Hoffmann, L., Holt, L. A., Alexander, M. J., Moffat-Griffin, T., and Mitchell, N. J.: Gravity waves
in the winter stratosphere over the Southern Ocean: high-resolution satellite observations and 3-D spectral analysis, Atmos. Chem. Phys.,
19, 15 377–15 414, https://doi.org/10.5194/acp-19-15377-2019, 2019.

Krisch, I., Ern, M., Hoffmann, L., Preusse, P., Strube, C., Ungermann, J., Woiwode, W., and Riese, M.: Superposition of gravity waves with
different propagation characteristics observed by airborne and space-borne infrared sounders, Atmos. Chem. Phys., 20, 11 469–11 490,
https://doi.org/10.5194/acp-20-11469-2020, 2020.

Lehmann, C. I., Kim, Y.-H., Preusse, P., Chun, H.-Y., Ern, M., and Kim, S.-Y.: Consistency between Fourier transform and small-volume
few-wave decomposition for spectral and spatial variability of gravity waves above a typhoon, Atmos. Meas. Tech., 5, 1637–1651, 430
https://doi.org/10.5194/amt-5-1637-2012, 2012.

Savitzky, A. and Golay, M. J. E.: Smoothing and Differentiation of Data by Simplified Least Squares Procedures., Analytical Chemistry, 36,
1627–1639, https://doi.org/10.1021/ac60214a047, 1964.

Wright, C. J., Hindley, N. P., Hoffmann, L., Alexander, M. J., and Mitchell, N. J.: Exploring gravity wave characteristics in 3-D using a novel
S-transform technique: AIRS/Aqua measurements over the Southern Andes and Drake Passage, Atmos. Chem. Phys., 17, 8553–8575,
https://doi.org/10.5194/acp-17-8553-2017, 2017

---

## Author Comment (AC2)

**Author response to Referee #2**

We would like to thank referee #2 for the valuable comments. We have included the comments one by one, in bold text, along with our answers. Lines in the answers refer to the original manuscript. The blue colour indicates text added in the revised manuscript.

**Specific comments**

- **Line 131: This study proposes to use the S3D spectral analysis method, but for completeness, the S-transform spectral method could be referenced as another commonly used method for analyzing the types of measurements to be provided by MATS.**

  Referee #1 also suggested mentioning alternative methods for scale separation and wave analysis. See response to referee #1.

- **Line 146: It is pointed out that the swath width of about 200 km of MATS is a bottleneck for properly estimating some of the gravity wave spectral characteristics. Why was the instrument built with such a small swath, assuming that the estimation of the gravity wave momentum flux in the MLT region is indeed a key objective of the MATS measurements?**

  When designing a telescope, one must compromise between the field of view and the resolution/sensitivity. MATS, which also considers smaller horizontal structures than those represented in the HIAMCM, was designed to prioritise the resolution, especially in the vertical - still weighing in a reasonably wide field of view. We have updated the manuscript with this information (Sect. 2.3 Orbit simulator):

  L121: Parameters were determined from preliminary orbit properties before MATS was launched. They vary slightly from the final orbit of MATS, which ended up at 17:30 local solar time (ascending node) and an altitude of 590 km. When designing a telescope, one must compromise between the field of view and the resolution/sensitivity. MATS, which also considers smaller horizontal structures than those represented in the HIAMCM, was designed to prioritise the resolution, especially in the vertical - still weighing in a reasonably wide field of view. The output from the MATS tomography may vary in resolution and dimension both geographically and in time, as the settings of the instrument may be controlled based on external factors.

- **Lines 148-150: Perhaps add the dimensions for clarity here, 600 km x 190 km x 10 km (along track x across track x vertical). It would be good to provide some rationale or reference as to why the vertical width of the S3D boxes was set to 10 km. It is pointed out that the length of the cuboids is 600 km, but the sampling would be done every 100 km. What would be the reason for this oversampling, since the results of neighboring boxes would be largely correlated?**

  The dimensions have been added in the updated manuscript (Sect. 2.4 S3D):

  L148: The cuboid sizes are therefore chosen to be 600 km x 190 km x 10 km (along-track x across-track x vertical).

The vertical extent of the cube is selected based on the results from (Chen et al., 2022), but we decided to test a slightly smaller cube size, as we are ultimately interested in the vertical variations of wave properties. This has been added to the revised manuscript:

L146: Because of the large horizontal wavelengths of the waves represented in HIAMCM, cuboid sizes should cover the entire across-track range while keeping a large along-track range. Vertically, the cuboid size is selected to be slightly smaller than what was used in Chen et al. (2022). With a smaller size, vertical variations of wave properties can be obtained at a higher resolution.

The answer to why we have overlapping cuboids is that a wave is not defined at a point, so it is hard to consider for example GWMF at a point. It is therefore natural to do the wave analysis in overlaps. The large cubes allow for characterisation of the waves but the overlaps allow us to see spatial changes at a higher resolution. This has been added to the manuscript:

L148: Each cuboid will be aligned with the centre of the across-track range of the data, with overlapping cuboids positioned every 100 km in the along-track direction. This does lead to large overlaps between cuboids along the track, but as waves are not defined at a point it is natural to analyze them in such a manner. The large along-track extent helps to correctly capture wave parameters while the overlap opens for the analysis of spatial changes at a reasonably high resolution.

- **Line 176: It is pointed out that the zonal wavenumber 18 corresponds to wavelengths of about 2200 km at the equator. If the reference method is to use zonal wavenumber 18 for scale separation, it looks like there would be some discrepancy with the measurements of the MATS instrument, which has a swath width of ~200 km, which in combination with the S3D could allow estimation of wavelengths up to ~600 km. Isn't there some kind of gap between 600 and 2200 km wavelength?**

ZWN decomposition is a common method for scale separation, despite that it implies a latitudinal dependence on the wavelength in the zonal direction. As the referee points out this means that the residual field close to the equator potentially includes waves that are much larger than the field of view.

The choice of our reference was based on a study by Strube et al., (2020), where it was shown that ZWN decomposition at ZWN 18 captures the significant momentum in the upper troposphere and lower stratosphere.

The good agreement between the GWMF derived from wind residuals and the GWMF derived from S3D (Figure 5) illustrates that the waves carrying significant GWMF were identified. There could indeed be waves with large zonal wavelengths in the residual fields that we couldn't characterize in the observational geometry, but as the GWMF agrees well, they either do not exist or do not carry significant momentum.

- **Lines 191-193: You might add that structured noise from something like stray light would also be very difficult and challenging to properly simulate and account for.**

  We have updated the manuscript with a note regarding this:

  L192: Structured noise, such as stray light, that could appear in the raw images taken by MATS, and then propagate into the tomography, is neglected. Such features are assumed to have been removed in the earlier stages of the data processing, as they are difficult to predict and properly account for.

- **Lines 240-241: I may have missed it earlier in the paper, but for completeness you might want to mention whether the HIAMCM data is a boreal summer and austral winter case?**

  This is mentioned in the article. On line 112 we specify that the HIAMCM snapshot is from January 1st, 2016. In the updated manuscript it is now also mentioned in the conclusions:

  L345: This study was performed on a HIAMCM snapshot from 1st January 2016.

- **Lines 269-271: The proposed cuboids have an asymmetry in size along and across the track, which seems to lead to an anisotropy in spectral sensitivity of the gravity wave measurements along and across the track. Such an anisotropy can cause some difficulty in interpreting measurement results. This can be seen in the example presented here for the polar regions. Perhaps you could discuss and clarify why using such unbalanced cube sizes is still advantageous compared to e.g. using a cube size of 200 km x 200 km x 10 km?**

  It is generally preferable to have cuboids with horizontal sides of equal lengths. The choice of non-equidistant cuboid size was selected based on the range limitations in the across-track dimension. Chen et al., (2022) successfully described the GWMF at 75 km using 600 x 600 cuboids and at 130 km using 300 x 300.

  For MATS observational geometry, the across-track is limiting, and along this dimension, we have made the cuboids as large as possible, i.e. 600 km x 200 km. This has the additional advantage that the number of points along each dimension is of similar size because of the different sampling rates across-track and along-track. This is now stated in the updated manuscript (Sect. 2.4 S3D):

  L148: The cuboid sizes are therefore chosen to be 600 km x 190 km x 10 km (along-track x across-track x vertical). This has the additional advantage that the number of points along the horizontal directions are of similar order because of the different sampling rates across-track and along-track: 31 x 39 x 11.

- **Lines 286-287: When discussing vertical smoothing, it would be good to provide some information on the vertical resolution of the HIAMCM test data. The MATS measurements have a vertical resolution of 500 m, and averaging**

**kernels with FWHMs of 1 km and 2 km are considered for the retrieval data. Is the HIAMCM vertical resolution comparable or better?**

The vertical resolution of HIAMCM is comparable, with a vertical resolution of approximately 600 m below 130 km. We add this to the discussions of lines 286-287.

L287: This is not surprising as the vertical resolution is close to the length scales of the vertical wavelengths in the data (HIAMCM approximately has a vertical resolution of 600 m below 130 km). Consequently, small changes can cut off significant amounts of momentum.

**Technical corrections**

- **line 201: Section -> Sect.**

  Addressed in the updated manuscript.

- **line 220: efficiently -> effectively (?)**

  Addressed in the updated manuscript.

**References**

Chen, Q., Ntokas, K., Linder, B., Krasauskas, L., Ern, M., Preusse, P., Ungermann, J., Becker, E., Kaufmann, M., and Riese, M.: Satellite $400$ observations of gravity wave momentum flux in the mesosphere and lower thermosphere (MLT): feasibility and requirements, Atmos. Meas. Tech., 15, 7071–7103, https://doi.org/10.5194/amt-15-7071-2022, 2022.

Strube, C., Ern, M., Preusse, P., and Riese, M.: Removing spurious inertial instability signals from gravity wave temperature perturbations using spectral filtering methods, Atmos. Meas. Tech., 13, 4927–4945, https://doi.org/10.5194/amt-13-4927-2020, 2020.